

# Head lice: impact of COVID-19 and slow recovery of prevalence in Cambridgeshire, UK

Ian F. Burgess, Elizabeth R. Brunton and Mark N. Burgess

Medical Entomology Centre, Insect Research & Development Limited, Cambridge, UK

## ABSTRACT

Following school closures and changes in contact behavior of children and adults a reduced head louse prevalence has been reported from across the globe. In parallel, sales of treatments were observed to fall, partly because of supply problems of some products following the pandemic, but this did not appear to result in more cases of infestation. Surveys of schools in and around Cambridge, UK, found that infestation rates were significantly reduced particularly in city schools compared with similar surveys conducted before the COVID-19 pandemic. Contrary to expectation the number of cases in schools has only risen slowly since schools returned to normal full time working in 2022–2023.

# INTRODUCTION

Head louse infestation in developed economies is principally found in children of primary school age extending into the first few years of secondary schooling, the latter often from contact with younger siblings. There are no data on transmissibility of head lice, but numerous point prevalence surveys have been conducted worldwide that show evidence of links between factors such as social crowding, forms of play, attendance at school, and out of school activities, with presence and transmission of head lice.

Irrespective of the actual point of transmission of lice, the regular attendance of children at school gives them the opportunity to encounter members of their peer group potentially carrying head lice that can be passed on. Similarly, attendance at more social groupings such as sports clubs, scout organizations, birthday parties, sleep overs, and selfie photographs increase the chances of acquiring head infestation. In addition, school holidays do not give respite from possible exposure because they may be associated with public holidays where friends and family gather, such as various winter and spring religious festivals, or family travels for vacations in the summer months where children enter play schemes and encounter those of a similar age from a variety of backgrounds and cultures.

These normal routines all became disrupted during the period from late 2019 through to mid-2021 as a result of government interventions attempting to control the transmission of SARS-CoV-2 (COVID-19) virus infection. This disruption included full lockdowns, school

Corresponding author
Ian F. Burgess,
ian@insectresearch.com

closures, and a slow return to normal processes of life, with various levels of intensity for each country. In the United Kingdom, some schools closed as early as late February 2020, with a nationwide shutdown of education from 20 March for all students except children of key workers, and vulnerable children. Although some schools started to reopen in June that year, the majority remained closed until the start of the new academic year in September 2020. Following a resurgence of virus transmission in December 2020 there was a further closure of schools from early January through until the beginning of March 2021 (*Wikipedia, 2023*).

Similar lockdown periods in other countries appear to have had an impact on transmission, detection, and putatively the elimination of head lice, which have been indicated through changes in the sales of head louse treatments in France, Israel, and in North Carolina, USA, (*Launay et al., 2022*; *Mumcuoglu, Hoffman & Schwartz, 2022*; *Bonanno, Lee & Sayed, 2022*) as well as responses to a questionnaire distributed online in Buenos Aires, Argentina (*Galassi et al., 2021*). For this article we have reviewed reported changes in sales figures of pediculicide products in the UK and Europe and looked at the presence of lice in schools before and after COVID-19 by combing examinations of children across whole schools.

## MATERIALS & METHODS

### Market data for head louse products

Sales data for the United Kingdom head louse market showing sales over 52-week periods from April 2016 through to March 2023 were generated from the market research company IRI. Data were originally obtained so that brands could be compared over time for market share. These include all products under the same label, including pharmacy sales and dispensing of medical products as well as pharmacy and supermarket sales of medical device products. Data showing annual product sales changes in European Union countries for 2019 to 2022 were kindly provided by Oystershell NV.

Data for prescribable medicines categorized by National Health Service geographic region of England were obtained from the OpenPrescribing website (*OpenPrescribing, 2023*). These show the number of prescriptions dispensed for each of the three head louse treatment medicinal products in the UK (dimeticone 4% cutaneous lotion or spray, malathion 0.5% aqueous liquid, permethrin 1% scalp application). Available searchable databases on the website include individual GP practices, local NHS trusts, or regional NHS teams.

### School screening

The Medical Entomology Centre has routinely worked with local schools throughout Cambridgeshire, UK, since 2006. Schools with a perceived or confirmed head louse problem have made contact on a regular basis to conduct whole school checks using dry detection combing using PDC plastic combs (Zantey ApS, Værløse, Denmark) under a continuing approval for combing and collecting lice granted by Huntingdon Research Ethics Committee (REC reference Number 06/Q010640). Usually, at the beginning of each school year, interested schools send a letter to parents/guardians requesting consent for

children to be checked on one or more occasions during the school year and to return a section of the letter if they do not wish their child(ren) to be checked. Only a small number of children's parents have declined each year. Children found to have lice are provided with a letter to take home and usually a voucher redeemable at a supermarket pharmacy local to the school. When appropriate, an offer to take part in a clinical investigation of a new treatment option is also given. Each voucher was valid for any head louse treatment product of the parent's choice to the value of £10.00 (approximately €12.00 /$12.00). This was generally sufficient to purchase one unit of some treatment products or most of the cost of more expensive products. Such screening visits were conducted regularly in several schools within a 30km radius of Cambridge from 2006 up until autumn 2019.

After schools' return to full time operation, and an expected recovery of head louse infestation in autumn 2022, a series of schools in Cambridgeshire were contacted with an offer to visit and screen the children in the same manner as previously.

## RESULTS

### Market data for products

In the UK sales of head louse treatment products had been in some decline since the introduction of physically acting products in 2006 and the withdrawal of several insecticide-based products. This decline continued through to the early part of 2019, nearly a whole year before the outbreak of COVID-19 (Fig. 1). The sequence of sales reports showed that the number of units of the main branded treatment products continued to decline during the period of the first lockdown. For example, sales of Brand 1 reduced from 949,953 units to 774,310 (175,643 units or 18.5%) and Brand 2 declined from 565,627 units to 389,863 (175,764 or 31.1%) between April 2019 and April 2020. However, products that had previously held a smaller market share, including minor brands, generics, and combs, increased their sales from 277,768 units to 1,283,760 over the same period, an increase in turnover of 462.2%. Essentially the same trend for downward sales of the main brands continued through to April 2021 with further losses of market share, with Brand 3 selling 191,794 (49.7%) fewer units (194,453 down from 386,247) and Brand 1 selling 432,772 (56.0%) units fewer (341,538 down from 774,310). For this year there was also a reduction in the "Others" category to 716,035 units, a reduction of 44.2% relative to 2019–2020 but this share was still more than 2.5 times higher than for 2018–2019 (Fig. 1).

A similar pattern emerged for sales of pediculicides throughout Europe (Fig. 2) with a decline in sales for most countries between the second quarters of 2019 and 2021, with a slight recovery in most countries during 2022. Between 2019 and 2022 the lowest decline was only 9% in Poland compared with the greatest decline of 59% in Italy. Overall, across the continental area the decline between 2019 and 2022 was 42% after a 17% recovery between 2021 and 2022.

In the UK, prescribed pediculicide treatments now constitute only a small fraction of treatments used since the changes of recommendations from the Department of Health (DoH) in the mid-1990s (*DoH, 1996*), after which many general practitioners either chose not to prescribe pediculicides or else strictly limited their prescriptions. However, three
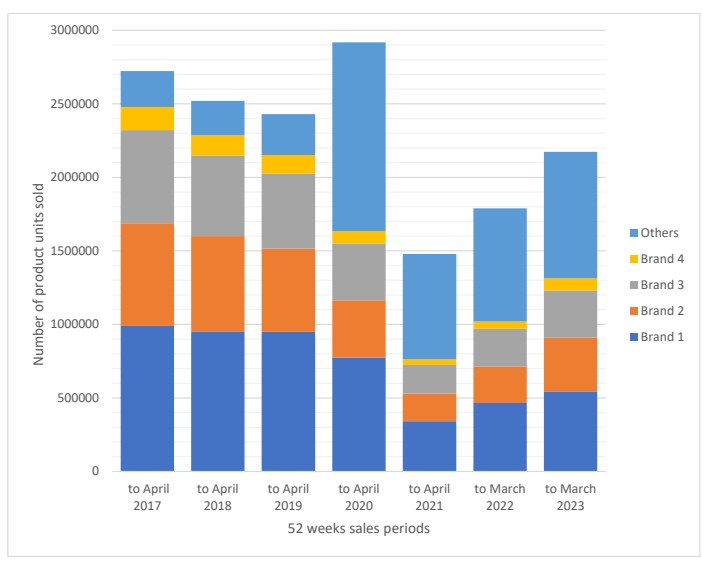

**Figure 1** **UK sales of head louse treatment products April 2016 to March 2023.** Each column shows the numbers of units of product sold for each of the four major brands, numbered "Brand 1" to "Brand 4", and the collective sales for minor brands, generics, and combs, shown as "Others".

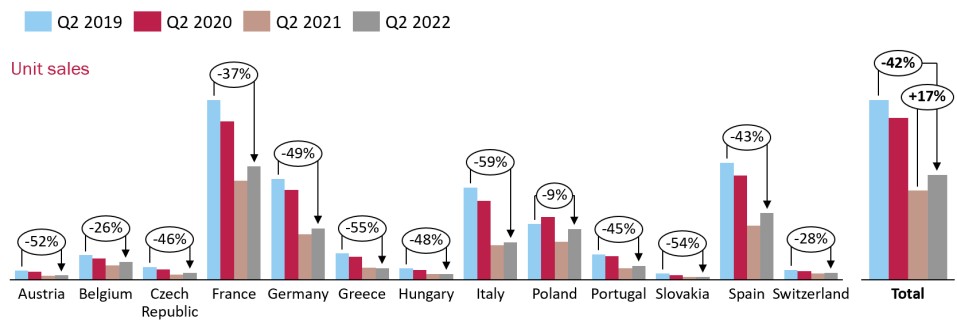

**Figure 2** **Sales of head louse treatment products in the European Union from June 2019 through to June 2022.** The columns show the relative numbers of units of product sold and the percentage change year on year.

medicinal treatment products are still available although only in a few of the Integrated Care Boards did the number of prescriptions each month reach 100 for any of the treatments even before 2019 but, as with over-the-counter sales of treatment products, there was generally a trend for a decline in prescribing between 2020 and 2022 (Fig. 3).

## School screening

During the lockdown periods the number of children in schools was only a fraction of normal, and varied considerably depending upon the location and the socio-economic structure of the communities they served. For example, in one local school between 80 and 100 children (20–25% of the school roll) still attended because their parents were mostly

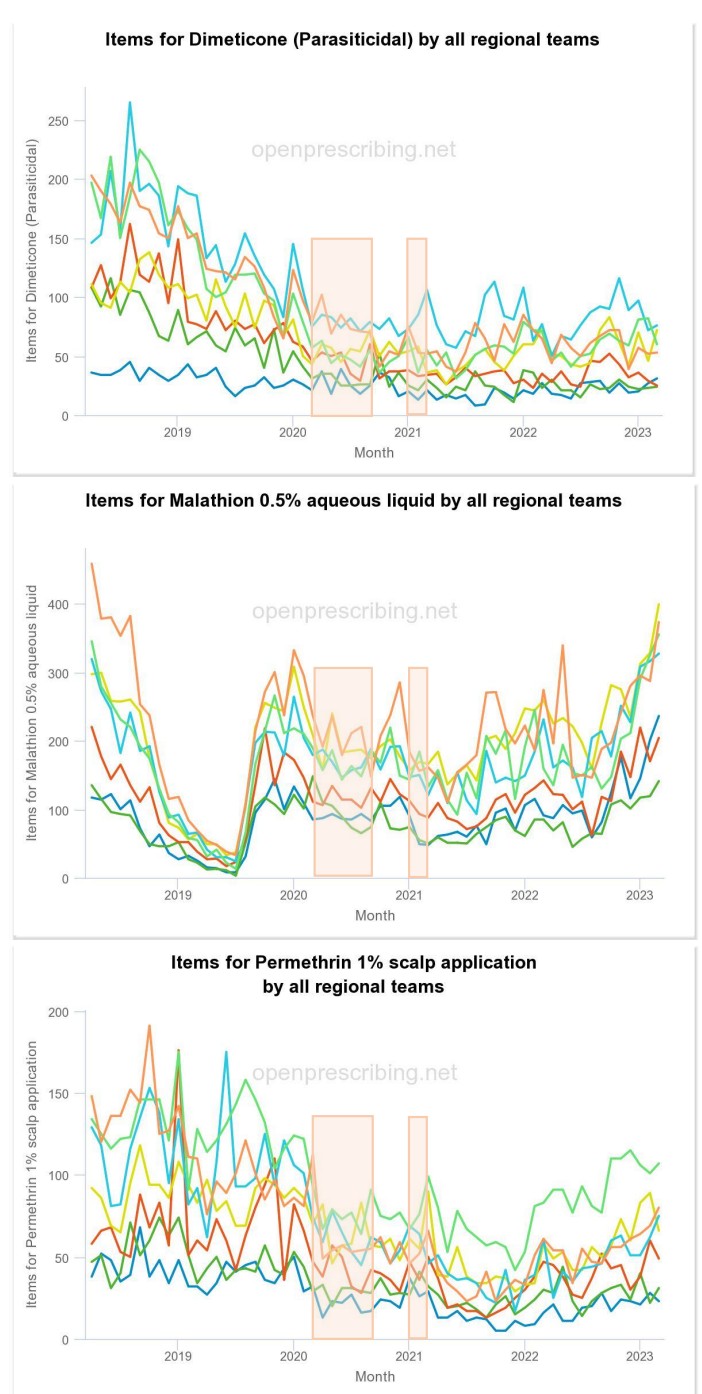

**Figure 3** NHS prescribing of insecticide products for head lice by geographic region 2018-June 2023.
Each colored line represents the prescribing in one National health Service region, however, the colors used for each region by the online graphic generator are not consistent from one product type to another, *e.g.*, the line for the East of England region is dark blue for dimeticone (parasiticidal) and dark green for malathion 0.5%. Note: the prescribing line of malathion 0.5% also includes prescriptions for scabies treatment. Periods of lockdown during the COVID-19 pandemic are shown by vertical bands of light orange color.

key workers whereas another school close by was only attended by children considered vulnerable or at risk, about the 2% of the roll. Despite government recommendations for "social distancing" it was clear that this meant little to primary school children and playground observations showed that most children played with as much contact as usual, which theoretically could have permitted normal levels of head louse transmission.

During the spring of 2022 we attempted to start recruitment for a clinical investigation of a head louse treatment using our normal method of advertising on local radio stations. The response was lower than expected compared with previous recruitment campaigns and suggested that perhaps the incidence of head louse infestation had not returned to normal.

By early September 2022, the beginning of the autumn term, it was expected that children playing together over the summer holiday would have enabled transmission of lice but, after contacting more than 30 of the schools previously visited, only four requested a visit. Some schools responded to follow up telephone calls by stating that they did not currently have a problem, so concluded because they were not receiving complaints about lice from parents. Of the four schools that did request a screening visit, schools B, F, G, and H, School B had been regularly visited previously since the mid-1990s (*Burgess, 2023*) and was considered a benchmark for the prevalence of lice in local communities. The combing examinations found highly significant differences ($p < 0.001$) in the levels of infestation for the two larger urban schools (B and H) compared with pre-COVID levels, shown in Table 1, but a non-significant reduction in city School G, which also had a considerably diminished school roll compared with previous visits. The rural School F was effectively unchanged from previous visits, mainly because of families with more than one child having lice together.

In January of 2023 it was expected that louse prevalence may have increased following family social gatherings during the winter holiday period. Rural School J was only marginally different from previous visits and, in early February, city School K, which also had a smaller school roll than on previous visits, appeared to indicate a slight recovery of lice in that community (Table 1). However, visits to a village school (P) and two city schools (Q and R) over the next 5 weeks showed that any "recovery" in louse transmission that had been observed in School K was highly local (Table 1).

A proportionately higher prevalence of 3.02% was found in May 2023, after the spring school holiday, in the urban School S, which had not been previously visited. A second visit to School K found no significant difference ($p = 0.846$) from February whereas in School B, visited 1 day later, there was a significantly ($p = 0.0278$) increased prevalence compared with the previous October. However, this prevalence of 2.89% was still significantly ($p = 0.0028$) lower than before the COVID pandemic (Table 1).

## DISCUSSION

From 2006 through to 2018 we found consistent levels of infestation when screening for head lice in primary schools throughout Cambridgeshire, Huntingdonshire, and adjacent areas. The most recent series of visits pre-COVID in 19 schools found 387/7169 (5.4%)

**Table 1  Presence of head lice in Cambridgeshire primary schools from October 2022 to June 2023 compared with pre-COVID prevalence in the same schools.** A statistical comparison was made between post-COVID and pre-COVID infestations using a 2-sided Fisher Exact Test calculator available at https://www.socscistatistics.com/tests/fisher/default2.aspx.

| School | Inspection for lice autumn 2022 | | | Pre-COVID inspection for lice | | | *p*-value |
|---|---|---|---|---|---|---|---|
| | Number of students | | | Number of students | | | |
| | Total | With lice | Percentage | Total | With lice | Percentage | |
| B Oct 3 | 420 | 3 | 0.71% | 425 | 33 | 7.76% | <0.001 |
| F Oct 12 | 99 | 6 | 6.06% | 112 | 6 | 5.36% | 1.00 |
| G Oct 18 | 188 | 4 | 2.13% | 305 | 15 | 4.92% | 0.150 |
| H Nov 29 | 320 | 5 | 1.56% | 318 | 32 | 10.06% | <0.001 |
| J Jan 30 | 97 | 2 | 2.06% | 105 | 4 | 3.81% | 0.684 |
| K Feb 6 | 302 | 13 | 4.31% | 540 | 31 | 5.74% | 0.422 |
| P Feb 22 | 297 | 5 | 1.68% | – | – | – | n/a |
| Q Mar 3 | 327 | 3 | 0.92% | 345 | 24 | 6.96% | <0.001 |
| R Mar 13 | 355 | 3 | 0.85% | 432 | 16 | 3.70% | 0.0095 |
| S May 22 | 398 | 12 | 3.02% | – | – | – | n/a |
| K Jun 7 | 297 | 14 | 4.71% | 540 | 31 | 5.74% | 0.632 |
| B Jun 8 | 381 | 11 | 2.89% | 425 | 33 | 7.76% | 0.0028 |

of students had lice, with a range across both rural and city schools of 1.35%–13.04% infestation, which was essentially similar to surveys conducted in the mid-1990s (*Burgess, 2023*). During this period, each child found to have lice was issued with a head louse treatment voucher but the number of vouchers returned by the pharmacies was consistently lower than the number issued and rarely exceeded 50% of those sent out from any of the schools, which suggests that either the parents chose not to purchase a treatment for their children or else chose alternative approaches to purchasing a product. As a result, head louse infestation, and presumably transmission, would have continued throughout each school year in many of the schools, as found previously (*Burgess, 2023*).

In this study's series of school screening visits, we have found that fewer students are attending the majority of schools we visited. This reduction in school rolls is partly a reflection of a declining birth rate both in the local area and nationally from 2010 through to 2021, with a 14% drop in Cambridge over the past decade (*United Kingdom Office for National Statistics, 2023*; *CfS, 2023*). This has been compounded by persistent absence by some students rounded up to about 25% of students during the autumn term of 2022 (*Thompson, 2023*). Although persistent absence is reportedly more likely for disadvantaged students, we saw no evidence that this may have contributed to lower detected rates of infestation. Cambridge has long had a different demography from the surrounding areas with a relatively cosmopolitan population. This series of school visits found that in the city, as opposed to surrounding villages, approximately half the students have family names that are not historically British in origin and reflect the current ethnic demographic. Cambridge currently has a population of which approximately 53% identify as white British (*Census, 2021*) and has the highest non-British European population of any place outside the inner London boroughs at around 20% (*Census, 2021*).

It could be postulated that households from different cultures have a different attitude to vigilance and management of head louse infestation for a variety of reasons so that the children who catch lice are diagnosed within the home and the infestation eliminated at an early stage. Such attitudes differ from those of some ethnic British households where lice can be found repeatedly on the children (*Burgess, 2023*; *Burgess & Brunton, 2023*) and from some of whom we have in the past heard comments such a "They should do something about it (lice)", the "They" being some unnamed and unidentifiable public authority, and which is presumably a hangover from the former practice of the school nursing services to supply pediculicides and to treat children, a practice discontinued over 30 years ago.

The impact of COVID-19 on levels of head louse infestation in other countries had until recently only been evaluated by comparisons of treatment product sales figures (*Launay et al., 2022*; *Mumcuoglu, Hoffman & Schwartz, 2022*; *Bonanno, Lee & Sayed, 2022*), questionnaire survey (*Galassi et al., 2021*), or monitoring of online searches in the UK for information on head lice (*Walker & Sulyok, 2023*) and not by physical screening of children. However, one recent study in Poland has compared pre- and post-COVID levels of head louse infestation in schools and kindergartens. Although the data from 28 schools and 64 kindergartens were merged as groups for analytical purposes, and cover ages 3–14, it appeared that there was a significant reduction of infestation post-COVID in both schools ($p = 0.0001$) and in kindergartens ($p = 0.0435$) (*Padzik et al., 2023*).

The figures we have cited from market research are confirmatory of the work of others but in addition we have been able to restart a school screening program that has provided additional confirmation of reduced levels of infestation, which would account for some diminished demand for treatment products. In contrast, anecdotal reporting in some public press media have suggested conflicting outcomes for head louse infestation as a result of the pandemic with parents in the UK (*Roberts, 2021*) and a commercial louse treatment service in the USA reporting just as many incidents of infestation as usual (*NBC News, 2022*). However, these may be the result of parent carers spending more time looking for lice and then doing something about them. There are also signs within the recent sales data that the return to more normal activities after the lockdown periods has resulted in either an increased incidence of head louse infestation in some areas, or at least an increased awareness of parents to look for the possibility that their children might contract a louse infestation.

An additional observation made during the period before a return to normal activity was that in many of the supermarkets and pharmacies in our locality the stocks of main brand head louse treatment products were reduced, with some products missing from shelves for several weeks at a time. Enquiries with manufacturers revealed that for some products there were supply chain issues that had resulted in reduced deliveries, which in turn would have resulted in lower sales figures across the industry. Under normal circumstances a reduction in availability of mainstream treatment products might have been expected to result in an increase in prevalence across the population if transmission of head lice was occurring at a normal rate. The fact that children in the schools we have visited have mostly shown a lower than normal point prevalence indicates that the transmission rate has been generally lower, either because prevalence was reduced to a minimal level at some point during the

COVID-19 pandemic or else, once children were cleared of their lice, there were either sufficient changes in their contact behaviors that lice were not able to find new hosts or else natural louse transmission rates are actually much slower than people have historically perceived them to be.

Of these two possible reasons for continued low prevalence the more likely is that the louse population was reduced to historically low levels during the 2019–2021 period and has not had the opportunity to recover. In the past we have routinely postulated that where children have regular social contact and prolonged periods of close play (*Burgess, 1995*), particularly for girls, lice would spread throughout the social group so that no one individual would have a heavy population of lice. In contrast, where individuals are isolated and unable to pass on their lice, we have found heavy infestations on those people (*Burgess, 2023*; *Burgess, Brown & Lee, 2005*).

## CONCLUSIONS

Reports from elsewhere have indicated that transmission of head lice among school children was reduced during the COVID-19 pandemic because of school closures and social distancing. During this period and following it, the sales of head louse treatment products were considerably reduced in both the United Kingdom and European Union, with only a slow recovery up until early 2023. We were able to confirm that the prevalence of head louse infestation in primary school aged children in Cambridge, UK, had been reduced following the pandemic by conducting detection combing visits to schools. The numbers of infestations found in 2022–23 was significantly reduced in most schools compared with visits to the same schools performed before 2019 and, although an increase in the number of cases was detected in a benchmark school between October 2022 and June 2023, the level of infestation was still significantly lower than before the COVID pandemic, indicating that the spread of lice in a school community is considerably slower than commonly believed.

## ACKNOWLEDGEMENTS

We wish to thank the staff of all the schools visited during the course of this study for their assistance during the combing screening visits. Data on sales of head louse treatment products in the United Kingdom were kindly supplied by Thornton & Ross Ltd, Huddersfield, UK, and data on sales in the European Union by Oystershell NV, Merelbeke, Belgium.

### Funding
The authors received no funding for this work.

## Competing Interests

The authors declare they have no competing interests. Ian F Burgess, Elizabeth R Brunton and Mark N Burgess are employed by the Medical Entomology Centre, Insect Research & Development Limited, an independent contract research consultancy.

## Author Contributions

- Ian F. Burgess conceived and designed the experiments, performed the experiments, analyzed the data, prepared figures and/or tables, authored or reviewed drafts of the article, and approved the final draft.
- Elizabeth R. Brunton conceived and designed the experiments, performed the experiments, authored or reviewed drafts of the article, and approved the final draft.
- Mark N. Burgess performed the experiments, authored or reviewed drafts of the article, and approved the final draft.

## Human Ethics

The following information was supplied relating to ethical approvals (i.e., approving body and any reference numbers):

Huntingdon Research Ethics Committee granted ethical approval for screening of children for head louse infestation and collection of lice in schools (REC reference Number 06/Q010640)

## Data Availability

All raw data, excluding the names of schools and names of children, are available in Table 1.

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
