# Peer review of "Head lice: impact of COVID-19 and slow recovery of prevalence in Cambridgeshire, UK"

_PeerJ, doi:10.7717/peerj.16001_

## Round 0.1 · original submission · Minor Revisions

Dear Dr. Burgess and colleagues:

Thanks for submitting your manuscript to PeerJ. I have now received two independent reviews of your work, and as you will see, the reviewers raised only a few minor concerns about the manuscript. Thus, these reviewers are optimistic about your work and the potential impact it will have on research studying the impact of COVID-19 on head lice infections. Thus, I encourage you to revise your manuscript, accordingly, taking into account all of the concerns raised by both reviewers.

The concerns of the reviewers are minor; thus, it should not take much effort to address these concerns to greatly improve your manuscript and ready it for publication.

Please note that reviewer 2 has included a marked-up version of your manuscript.

I look forward to seeing your revision, and thanks again for submitting your work to PeerJ.

Good luck with your revision,

Best,

-joe

·

Basic reporting

This study is one of the earliest and most comprehensive ones showing how the head lice contagion evolved over different periods of time, where the spread was affected by the lack of contact among its hosts, a crucial factor for the disease transmission.

Experimental design

It clearly explains, through various methodologies (such as school surveys and product sales), how the infestation varied.
It is also important to mention that the authors did an excellent job differentiating between schools in towns and cities and comparing the situation in both, leading to realistic conclusions.

Validity of the findings

The study is very well organized, and the only point to mention is that Figure 1 should display percentages, as they are discussed in the main body of the work. Referring to the figure in the text and then reviewing it can cause some confusion in reading, as the vertical axis mentions the number of products sold.

Additionally, the percentage value in line 111 should be corrected.

I would like to congratulate the authors once again for this study.

Reviewer 2 ·

Basic reporting

The work covered the impact on the sale of pediculicides before, during and after COVID-19 pandemic in UK. Moreover, the authors reported evidence on the sale market share of several countries of Europe. English language is correct due to their authors are from UK. Literature cites are updated and mentioned the most relevant works related to the studied topic. Table and figures are well structured and they were clear and easy to read. The main expectation of the studied relied in the sale of OTC products to treat pediculosis before and after COVID-19.

Experimental design

The Methodology employed by the authors was clear, relevant and followed all the scientific procedures to cover all the proposed objectives. Also, the article reviewed reported changes in sales figures of pediculicide products in the UK and Europe and studied head lice prevalence before and after Covid-19 by combing examinations of children across whole schools. Authors declared to have all the right permits to design this study.

Validity of the findings

The work reported for the first time the sales of treatments against head lice before and after the COVID-19 in UK. It also added relevant information about the sales in 13 European countries. Finally, authors showed surveys of schools in and around Cambridge, UK, and found that infestation rates were significantly reduced particularly in city schools compared with similar surveys conducted before the Covid-19 pandemic. They compare their results with similar works indicating that transmission of head lice among school children was reduced during the Covid-19 pandemic because of school closures and social distancing.

Annotated reviews are not available for download in order to protect the identity of reviewers who chose to remain anonymous.

---

## Round 0.2 · accepted · Accept

Dear Dr. Burgess and colleagues:

Thanks for revising your manuscript based on the concerns raised by the reviewers. I now believe that your manuscript is suitable for publication. Congratulations! I look forward to seeing this work in print, and I anticipate it being an important resource for groups studying the impact of COVID-19 on head lice infections. Thanks again for choosing PeerJ to publish such important work.

Best,

-joe